# Online Formative Assessment in Higher Education: Bibliometric Analysis

**Natalia E. Sudakova** [1], **Tatyana N. Savina** [2], **Alfiya R. Masalimova** [3,*], **Mikhail N. Mikhaylovsky** [4], **Lyudmila G. Karandeeva** [5] **and Sergei P. Zhdanov** [6,7]

1　UNESCO, Russian Presidential Academy of National Economy and Public Administration (RANEPA), 119571 Moscow, Russia; sudakovane@bk.ru
2　Department of Theoretical Economics and Economic Security, Ogarev Mordovia State University, 430005 Saransk, Russia; savinatn@yandex.ru
3　Department of Pedagogy of Higher Education, Kazan (Volga Region) Federal University, 420008 Kazan, Russia
4　Department of Nursing Activities and Social Work, I.M. Sechenov First Moscow State Medical University (Sechenov University), 119435 Moscow, Russia; minih200707@yandex.ru
5　Department of Theory and Practice of Foreign Languages, Peoples' Friendship University of Russia (RUDN University), 117198 Moscow, Russia; karandeeva-lg@rudn.ru
6　Department of Civil Law Disciplines, Plekhanov Russian University of Economics, 115093 Moscow, Russia; zhdanov120009@yandex.ru
7　Department of Customs Law and Customs Organization, Russian University of Transport, 127055 Moscow, Russia
*　Correspondence: alfkazan@mail.ru

**Abstract:** Assessment is critical in postsecondary education, as it is at all levels. Assessments are classified into four types: diagnostic, summative, evaluative, and formative. Recent trends in assessment have migrated away from summative to formative evaluations. Formative evaluations help students develop expertise and concentrate their schedules, ease student anxiety, instill a feeling of ownership in students as they go, and confirm the module's subject notion. Online formative assessment (OFA) emerged as a result of the convergence of formative and computer-assisted assessment research. Bibliometric analyses provide readers with a comprehensive understanding of a study topic across a particular time period. We used a PRISMA-compliant bibliometric method. The Scopus database was searched for BibTex-formatted publication data. In total, 898 studies were analyzed. According to the results, *Assessment & Evaluation in Higher Education* and *Computers & Education* are the most influential sources. RWTH Aachen University and Universitat Oberta De Catalunya are the most effective institutions. The red cluster includes terms associated with higher education and evaluation. The word "e-assessment, e-learning, assessment, moodle" appears in the green cluster. This group is quite influential yet has a low centrality. The highest percentage is 79.2 for "online assessment". The subject is comprised of three components: "distance learning", "accessibility", and "assessment design". The most important topics were "e-assessment", "higher education", and "online learning". According to the country participation network, the USA and UK were the two main centers.

**Keywords:** online formative assessment; higher education; bibliometrics analysis

## 1. Introduction

Although it is perceived as an issue because assessment is defined differently for different procedures and purposes [1] it is self-evident that assessment has a substantial impact on learning. Assessment is essential in higher education, just as it is vital at all educational levels [2]. In fact, as stated by Bransford et al. [3], assessment is a core component for effective learning. There are four types of assessments: diagnostic; summative; evaluative; and formative. Formative vs. summative were defined primarily in terms of their goal and timing: (a) formative, in order to recognize and discuss a student's accomplishments

and arrange the necessary next steps; and (b) summative, for the methodical recording of a student's overall performance [4].

For the purpose of providing students with feedback that might help them learn and teach better, formative assessment is utilized [5]. During the development of teaching, formative assessment can also be referred to as assessment for learning [6–9]. There is no doubt that formative assessment has its advantages, and studies have demonstrated that these methods help students attain higher academic goals [10–12]. Recent assessment trends have shifted away from summative assessments, in which students' achievements are checked, and toward formative assessments, in which the assessment is utilized for learning and used in learning [13]. Formative assessment is a type of assessment used to provide students with feedback while they are learning and to enhance the curriculum and teaching techniques [7,14].

The assessment of learning by summative assessment has taken a back seat to the assessment of learning by formative assessment in assessment circles. However, the emphasis has moved dramatically due to the widespread use of online and blended learning in higher education in the twenty-first century [15]. Assessment in online learning environments covers various aspects as opposed to face-to-face situations, primarily due to the asynchronous nature of engagement among the online participants. Therefore, it demands educators to rethink online pedagogy in order to accomplish successful formative assessment procedures that can promote meaningful learning and its assessment [6,16].

Formative assessment activities are ingrained in guidelines for monitoring learning and assessing learners' comprehension in order to adapt instruction and influence additional learning through continuous and timely feedback until the desired level of understanding is attained [17]. Formative assessments are practical in that they enhance expertise and focus scheduling, alleviate student anxiety, provide students with an added sense of ownership as they advance, and, ultimately, validate the module's content idea [18–21].

Due to the advent of technology and, at times, need, education has shifted to an online format. Numerous nations have been forced to transition to online and distant education at various levels of education as a result of the current COVID-19 outbreak [22,23]. Assessment has also benefited from this change, particularly formative assessment, which has evolved into online formative assessment.

Online formative assessment (OFA) evolved as a result of a convergence of research in formative assessment and computer-assisted assessment. Prior assessments of the literature on formative and computer-assisted assessment consolidated essential information in these two domains of study [24–30].

In order to develop learner and assessment-centered learning environments, Pachler et al. [31] and Wang et al. [32] advised a refocused emphasis on OFA. However, according to Gikandi et al. [33], a search of the literature revealed no study of OFA. There are literature studies (such as Gikandi et al. [33]) on the use of OFA in higher education. However, no bibliometric study and scientific mapping examining the use of OFA in higher education has been found. According to Thanuskodi [34], Pritchard was the first to create the word "Bibliometrics" in 1969. Bibliometric methods of assessment are used by researchers to determine the impact of a single author or to define the link between two or more authors or works. Another approach in evaluative bibliometrics is science mapping, which aims to highlight structural and dynamic characteristics of scientific research [35]. This study aimed to conduct scientific mapping and bibliometric analysis related to the studies on the use of OFA in higher education.

1.  Which are the most relevant and cited studies, authors, affiliations, and sources relating to online formative assessment?
2.  What are the trend topics in online formative assessment?
3.  What are the research themes in online formative assessment?
4.  What are co-occurrence, co-citation, and countries' collaboration?

## 2. Method

Bibliometric analyses enable readers to gain a holistic view of the chosen topic of research throughout a specified time period [36]. We employed a bibliometric technique that adhered to the PRISMA guidelines [37]. The Scopus database was used to find relevant studies.

### 2.1. Data Sources

The Scopus is a world-class research platform that enables the discovery, analysis, and sharing of knowledge in the sciences, social sciences, arts, and humanities. The Scopus database contributes to the efficiency and effectiveness of the research workflow [38]. The Scopus database was preferred because it indexes the leading journals in the field of education and provides appropriate data for bibliometric analysis. The Scopus database was used to find relevant studies. Different keywords were preferred, and the most comprehensive search was performed. An online search was performed on the Scopus database website. "online formative", "e-assessment", and "higher education" were chosen as search keywords. Then, some restrictions such as language were applied. Finally, the following search term was applied.

TITLE-ABS-KEY ((*"online assessment"* OR *"online formative assessment"* OR *"alternative assessment"* OR *"e-assessment"*) AND (*"higher education"* OR *"university"*)) AND (LIMIT-TO (LANGUAGE, *"English"*)) AND (EXCLUDE (PUBYEAR, *2022*)).

Because the publications for 2022 were not completed, they were omitted from the scope of the searches. The requirement that the publication be in English has been added. In total, 927 publications were discovered as a result of the search (as shown in Figure 1). The Scopus database was queried for publication data in BibTex format. To begin, articles without author or publication year were eliminated from the data. Later, it was investigated for repeated articles, and the others were removed, leaving only one publication. There were various instances in book publishing, depending on the volume of books published each year. New publications were deleted based on the citation information. If a publication was simultaneously published as a congress paper and an article, the article was preferred. Finally, by studying the titles, it was determined whether there were any publications that were unconnected. At the conclusion of the exclusion, there were 898 broadcasts.

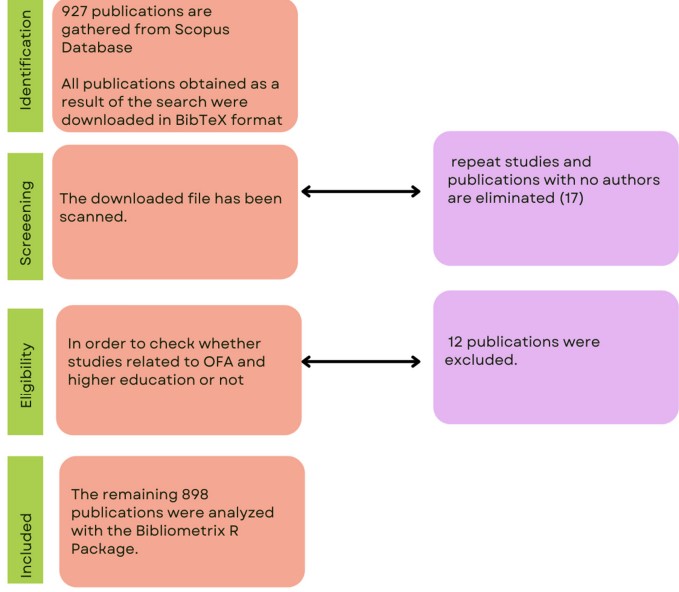

**Figure 1.** Flow Diagram for Bibliometrics Search.

### 2.2. Data Analysis

The present bibliometric study was conducted using the shiny app for bibliometrix from the R Statistical Package. It offers a number of characteristics that make it ideal for

doing in-depth bibliometric analysis. It is a web-based program that allows access to bibliometrix 3.1.4 capabilities [39]. Finally, the Scopus file was submitted to the Biblioshiny interface in BibTeX format.

In bibliometric analysis, there are two main parts. The first one is descriptive and performance analysis. In this analysis, there was general information on sources and document types. Additionally, statistical information on annual and total number of studies and citations was calculated. Then, the most cited studies were presented based on top 10 or 20 studies. Finally, the most productive authors, sources, institutions, or counties were introduced. The second analyses were based on scientific mapping and network analyses. Clusters by document coupling were analyzed based on authors' keywords. Thematic maps of online formative assessments are presented. Co-occurrences network, co-citations network, and country participations were analyzed based on a network approach.

## 3. Results and Discussion

In total, 898 studies were analyzed (As shown in Table 1). The studies covered the years 1998 through 2021. The papers were compiled from 556 different sources. These studies comprised 504 journal articles (56.1%), 303 conference papers (33.7%), 56 book chapters (6.2%), 21 reviews (2.3%), and 14 other documents (book, note, editorial, erratum). Most of the studies were journal articles. The indexing of academic journals in the searched database may have had an effect on this result. The studies included 3029 indexed keywords and 2049 author's keywords. The studies had 2351 authors and 2671 author appearances. The number of authors of single-authored documents was 168, and the number of authors of multi-authored documents was 2183. When the documents were examined in terms of authors' collaborations, there were 174 single-authored documents. There were 0.382 documents per author, and there were 2.62 authors per document. There were 2.97 co-authors per documents, and the collaboration index was 3.02.

**Table 1.** Descriptive Information.

| Description | Results |
| --- | --- |
| **Main Information about Data** | |
| Timespan | 1998:2021 |
| Sources (Journals, Books, etc.) | 556 |
| Documents | 898 |
| Average years from publication | 7.22 |
| Average citations per documents | 8.739 |
| Average citations per year per doc | 1.021 |
| References | 27,114 |
| **Document Types** | |
| Article | 504 |
| Conference paper | 303 |
| Book chapter | 56 |
| Review | 21 |
| Book | 9 |
| Note | 3 |
| Erratum | 1 |
| Editorial | 1 |
| **Document Contents** | |
| Keywords Plus (ID) | 3029 |
| Author's Keywords (DE) | 2049 |
| **Authors** | |
| Authors | 2351 |
| Author Appearances | 2671 |
| Authors of single-authored documents | 168 |
| Authors of multi-authored documents | 2183 |
| **Authors Collaboration** | |
| Single-authored documents | 174 |
| Documents per Author | 0.382 |
| Authors per Document | 2.62 |
| Co-Authors per Documents | 2.97 |
| Collaboration Index | 3.02 |

Figure 2 contains data on the annual scientific production of OFA. Sharp rises in this graph indicate a relatively higher increase in the number of publications in that year. There has been an increase in publications over the years. The annual growth rate is 17.44%. The rate of increase between 2015 and 2017 is lower than other years. The rate of increase between the years 2019–2021 is higher than in other years. Compulsory distance education studies due to COVID may have been effective in the increase in the number of studies.

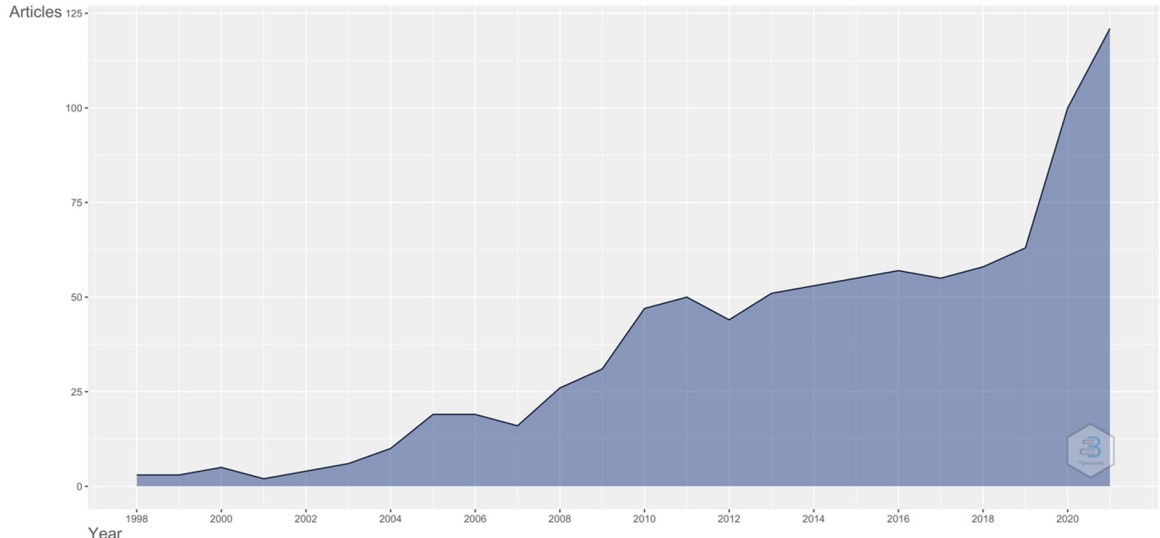

**Figure 2.** Annual scientific production of OFA.

In Figure 3, the variation of the total citation average over the years and the comparison of the average citations per study are given. Since the number of citations of the articles published in recent years will be less than in the previous publications, it is natural that the difference between the two graphics has increased in recent years. When the number of citations by years is examined, the citation rate in 2002 covers 20 years and received an average of 51 citations per document. The average number of citations per documents by year was 2.56. The average number of citations per document has decreased over the years because more recent publications have fewer years in which to be cited. The citation average over the years was 1.21.

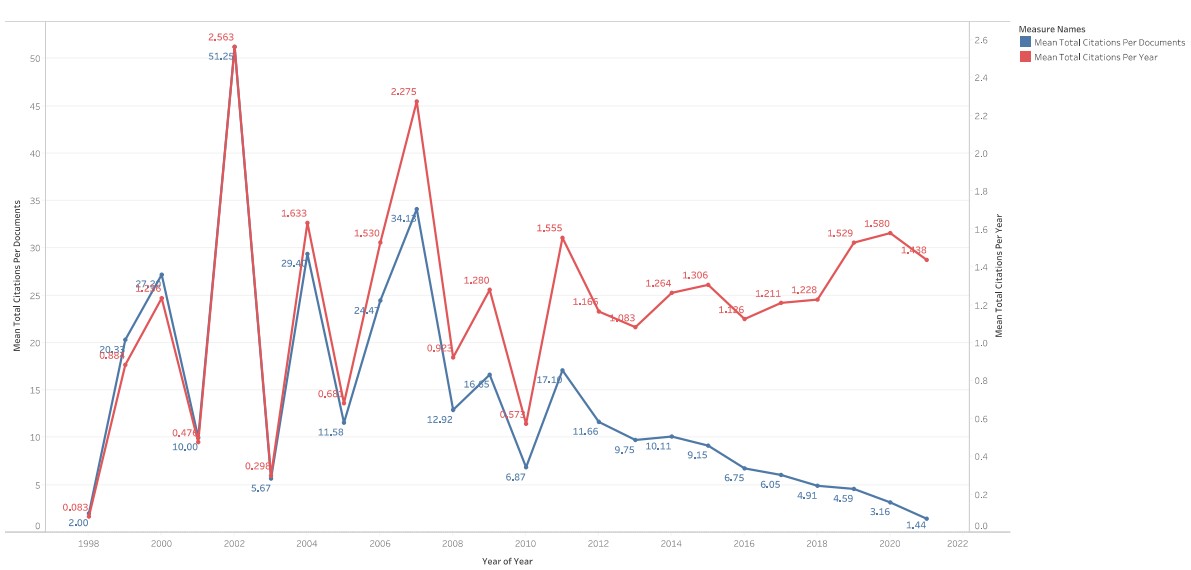

**Figure 3.** Annual citation per document over years.

As shown in Table 2, the studies obtained, the *Assessment and Evaluation in Higher Education* journal comes to the fore with 649 citations when the sources that contribute more are examined. Then, the *Computers & Education* journal comes to the fore with 543 citations. The least cited journal in the top 20 list was *Assessment in Education*, with 53 citations.

**Table 2.** Source Impact.

| Sources | Articles |
|---|---|
| Assessment & Evaluation in Higher Education | 649 |
| Computers & Education | 543 |
| British Journal of Educational Technology | 245 |
| Studies in Higher Education | 193 |
| Higher Education | 119 |
| Computers in Human Behavior | 100 |
| Review of Educational Research | 97 |
| Comput Educ | 89 |
| Journal of Computer Assisted Learning | 83 |
| Studies in Educational Evaluation | 79 |
| The Internet and Higher Education | 79 |
| Med Teach | 78 |
| Teaching in Higher Education | 70 |
| Language Testing | 65 |
| Assessment in Education: Principles | 63 |
| Australasian Journal of Educational Technology | 60 |
| Educational Technology Research and Development | 57 |
| Journal of Educational Psychology | 57 |
| Educational Researcher | 55 |
| Assessment in Education | 53 |

The annual increases of five journals with high impact values are examined in Figure 4. The increase in the graph shows that the effect of the journal is increasing gradually. When total citations per year is analyzed, the *Assessment and Evaluation in Higher Education* journal citations are increased over years. The journal *Communications in Computer and Information Science* began in 2011, and last year, it was the second-most cited journal.

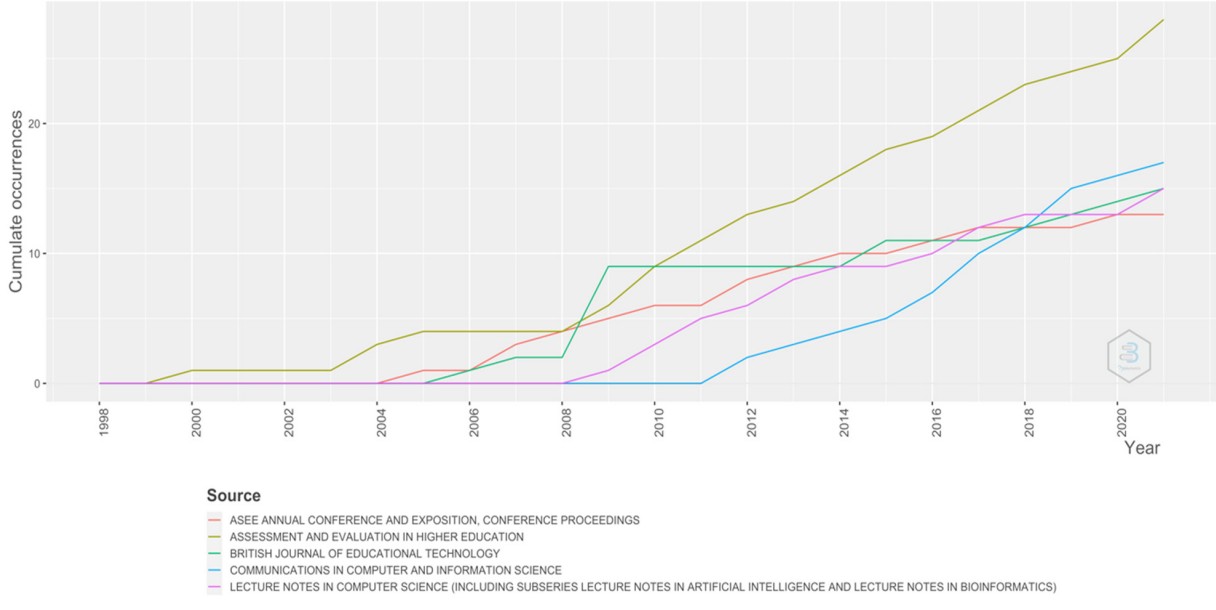

**Figure 4.** The Impact of first 5 source over years.

The top articles according to citations are listed in Table 3.

**Table 3.** Impact of studies based on citations.

| Paper | DOI | Total Citations | TC Rank | TC per Year | TC per Year Rank | Normalized TC | Normalized TC Rank |
|---|---|---|---|---|---|---|---|
| Gikandi et al. [33] | 10.1016/j.compedu.2011.06.004 | 426 | 1 | 35.5 | 1 | 24.9 | 2 |
| Christensen et al. [40] | 10.2196/jmir.4.1.e3 | 197 | 2 | 9.4 | 12 | 3.8 | 26 |
| Nicol [27] | 10.1080/03098770601167922 | 184 | 3 | 11.5 | 5 | 5.4 | 21 |
| Tsai et al. [41] | 10.1016/j.compedu.2011.07.012 | 167 | 4 | 15.2 | 3 | 14.3 | 3 |
| Ivanitskaya et al. [42] | 10.2196/jmir.8.2.e6 | 163 | 5 | 9.59 | 10 | 6.7 | 18 |
| Vallejo et al. [43] | 10.2196/jmir.9.1.e2 | 146 | 6 | 9.1 | 13 | 4.3 | 25 |
| Condon and William [44] | 10.1016/j.intell.2014.01.004 | 130 | 7 | 14.4 | 4 | 12.9 | 5 |
| Vonderwell et al. [6] | 10.1080/15391523.2007.10782485 | 125 | 8 | 7.8 | 18 | 3.7 | 27 |
| Kandiah et al. [45] | 10.1016/j.nutres.2005.11.010 | 124 | 9 | 7.3 | 20 | 5.1 | 22 |
| Neighbors et al. [46] | 10.1037/0893-164X.22.3.433 | 118 | 10 | 7.8 | 16 | 9.1 | 11 |
| Dermo [47] | 10.1111/j.1467-8535.2008.00915.x | 110 | 11 | 7.9 | 17 | 6.6 | 19 |
| Higgins et al. [48] | 10.1145/1163405.1163410 | 107 | 12 | 5.9 | 24 | 9.3 | 9 |
| Howley et al. [49] | 10.1097/00001888-200403000-00017 | 100 | 13 | 5.2 | 27 | 3.4 | 29 |
| Reeves [50] | 10.2190/GYMQ-78FA-WMTX-J06C | 94 | 14 | 4.1 | 30 | 3.5 | 28 |
| Frank and Barzilia [51] | 10.1080/0260293042000160401 | 91 | 15 | 4.8 | 28 | 3.1 | 30 |
| Dvorak et al. [52] | 10.1016/j.jad.2013.01.046 | 82 | 16 | 8.2 | 14 | 8.4 | 12 |
| Angus and Watson [2] | 10.1111/j.1467-8535.2008.00916.x | 78 | 17 | 5.6 | 25 | 4.7 | 23 |
| Stödberg [53] | 10.1080/02602938.2011.557496 | 76 | 18 | 6.9 | 21 | 6.5 | 20 |
| Draper [54] | 10.1111/j.1467-8535.2008.00920.x | 75 | 19 | 5.4 | 26 | 4.5 | 24 |
| Llamas-Nistal et al. [55] | 10.1016/j.compedu.2012.10.021 | 69 | 20 | 6.9 | 22 | 7.1 | 16 |
| Garcia-Penalvo et al. [56] | 10.1007/978-981-15-7869-4_6 | 40 | 23 | 20 | 2 | 27.8 | 1 |

When we examined the most cited studies, the study by Gikandi et al. [33] ranked first with 426 citations. Although it was in first place for the average number of citations per year, its ranking decreased to second when the normalization process was performed. According to Agarwal et al. [57], review articles are more likely to be cited than research articles. Similarly, when author effectiveness is examined, a similar situation arises when only the total number of citations is taken into account. The study with the second largest number of citations was Christensen et al. [40], with 197 citations. When it was normalized, it regressed to the 12th rank. According to Agarwal et al. [57], the use of normalized citation numbers somewhat compensates for the comparisons that take into account the total citation numbers. When the rank is changed according to normalization by years, García-Peñalvo et al. [56] will be in first rank, with a score of 27.8. The total number of citations for this study was 40. This study focused on theoretical and practical applications related to online assessment used in compulsory distance education due to the COVID-19 pandemic.

Top authors based on rankings according to the total citations and indexes received by the authors are listed in Table 4. When the contribution of authors was analyzed in terms of first three places of h-index, g-index, m index, and total citations, the authors in

the first three places according to TC do not rank higher than 15th in other indexes. The h-index is based on the number of citations that an author's most referenced published works have achieved in other publications. The h-index is also discipline-specific, with highly specialized academic scholars receiving fewer citations due to a smaller audience, resulting in a low h-index [58]. The g-index is the biggest (unique) number, such that the top g articles earned at least $g^2$ citations (all together) [59]. M-index considers the citing paper's dependability and the type of polarity between the citing and cited papers. Unlike the h-index, which is more author-specific, this index focuses primarily on a single publication [60]. According to m-index, the first-ranking author (Garca-Pealvo Fj) has 42 total citations, and their publication year start was 2021. The second-ranking authors are Al Abdulmonem W and El Sadik A.

**Table 4.** Impact of authors based on citations.

| Authors | h Index | h Rank | g Index | g Rank | m Index | m Rank | TC | TC Rank | NP | PY Start |
|---|---|---|---|---|---|---|---|---|---|---|
| Gikandi Jw | 2 | 17 | 2 | 22 | 0.167 | 42 | 432 | 1 | 2 | 2011 |
| Davis Ne | 1 | 36 | 1 | 36 | 0.083 | 53 | 426 | 2 | 1 | 2011 |
| Morrow D | 1 | 37 | 1 | 37 | 0.083 | 54 | 426 | 3 | 1 | 2011 |
| Velan Gm | 4 | 1 | 4 | 8 | 0.27 | 34 | 72 | 23 | 4 | 2008 |
| Ibarra-Siz Ms | 4 | 2 | 6 | 2 | 0.40 | 22 | 46 | 31 | 6 | 2013 |
| Guerrero-Roldn Ae | 3 | 3 | 6 | 1 | 0.33 | 28 | 75 | 22 | 6 | 2014 |
| Rodrguez-Gmez G | 3 | 7 | 5 | 3 | 0.30 | 29 | 42 | 32 | 5 | 2013 |
| Garca-Pealvo Fj | 2 | 25 | 2 | 29 | 1 | 1 | 42 | 33 | 2 | 2021 |
| Al Abdulmonem W | 2 | 22 | 2 | 26 | 0.667 | 2 | 57 | 28 | 2 | 2020 |
| El Sadik A | 2 | 23 | 2 | 27 | 0.667 | 3 | 57 | 29 | 2 | 2020 |

Note: TC: total citations NP: number of publications PY: publication year start.

In the Figure 5, the changes in the publication and citation status of the authors by year are presented. Red lines indicate continued citation. Whereas the size of the circular parts indicates the number of publications, the increase in the darkness of the blue color indicates the number of publications in that year. When the productivity of the authors over time is examined, Schroeder U has 10 years of production, from 2010 to 2019. Then, Babo R and Garca-Pealvo FJ have 9 years. Two of the most outstanding authors were Ibarra-Siz MS and Rodrguez-Gmez G. They only published between 2013 and 2016, but they were included in the top 10 list due to their high number of citations and number of publications. Corresponding Author's Country distrubiton is presented in Figure 6. The authors' nationalities were distributed among 20 countries. The nmajor countries were the USA, UK, Australia, Spain and Germanny (Figure 6).

The total publications of the institutions are charted in Figure 7. When the effects of the institutions are examined, the most effective ones are RWTH Aachen University and Universitat Oberta De Catalunya. Then, it is Qassim University and University of New South Wales, with 12 publications. Griffith University, Monash University, Open University, Technical University of Sofia, and Universiti Teknologi Malaysia are at the end of the top 20 list, with 6 publications.

A trend topic is created based on the frequent use of keywords (As shown in Figure 8). If a term is used with a certain frequency in the specified year, the blue line continues. The size of the flats shows the total usage size in that year. When looking at which themes were more popular over time, the phrases "blended learning" rose to the forefront between 2010 and 2020, inclusive. "Formative assessment" and "assessment" keywords are used high frequently between 2011 and 2019. The authors frequently added "moodle" to their keywords between 2015 and 2019. It is seen that "COVID" has started to be used intensively in publications in 2020 as keywords.

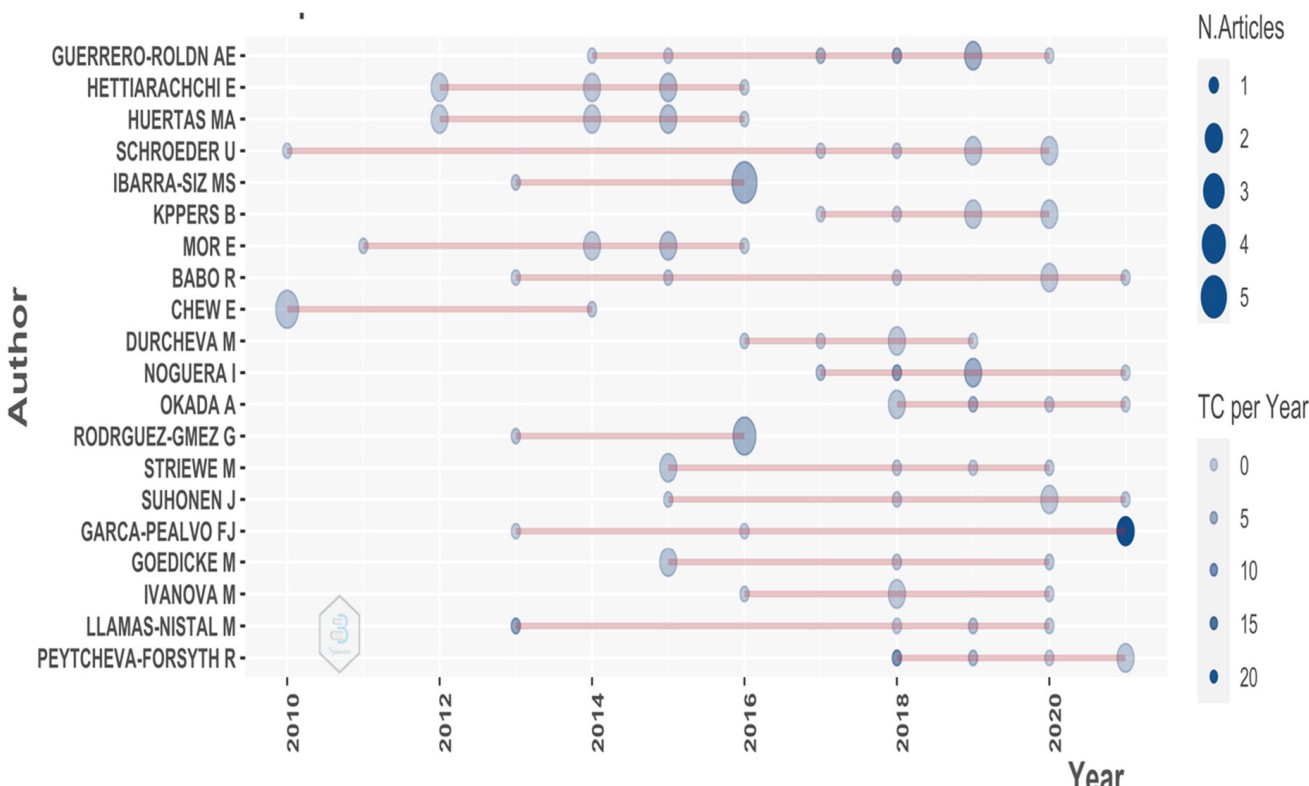

**Figure 5.** Top authors' production over the time.

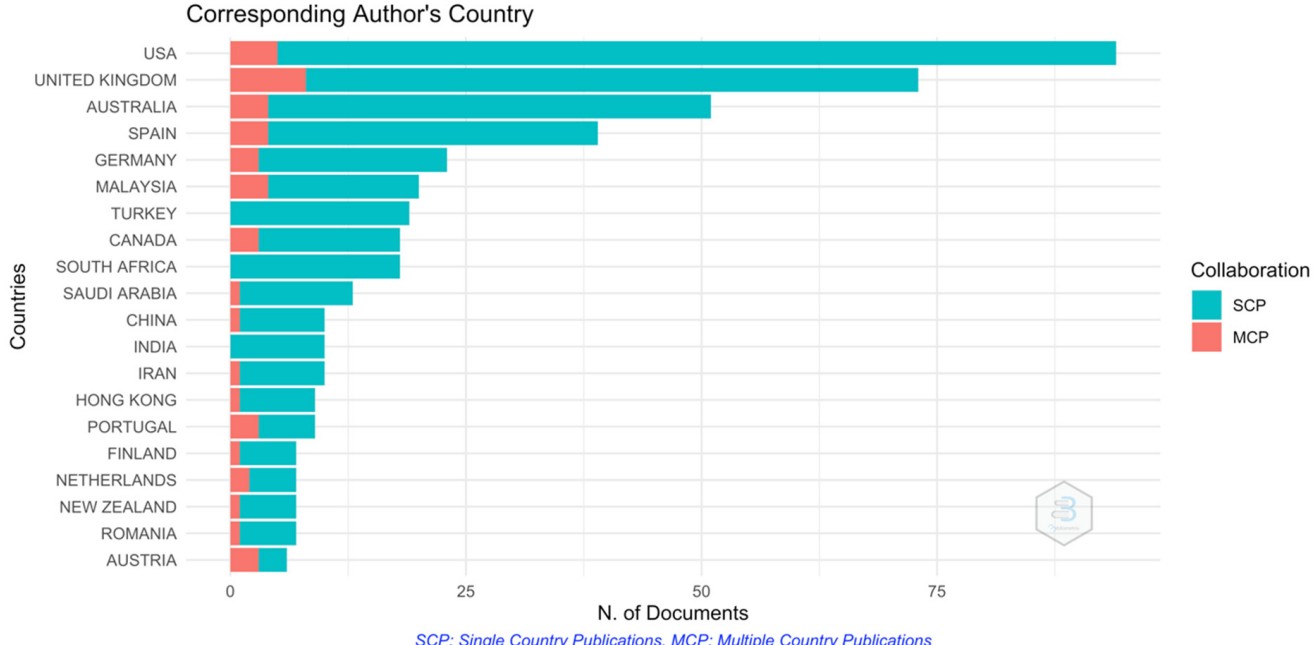

SCP: Single Country Publications, MCP: Multiple Country Publications

**Figure 6.** Corresponding Author's Country.

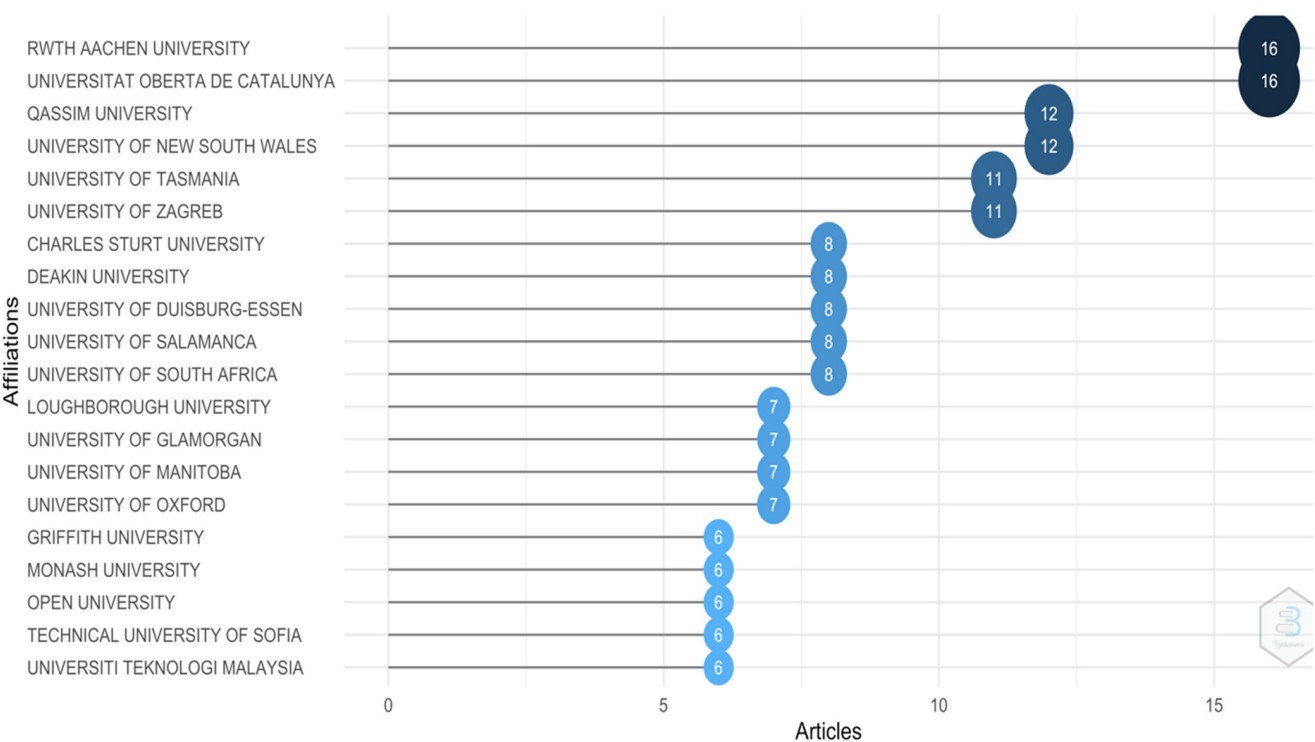

**Figure 7.** Most relevant affiliations.

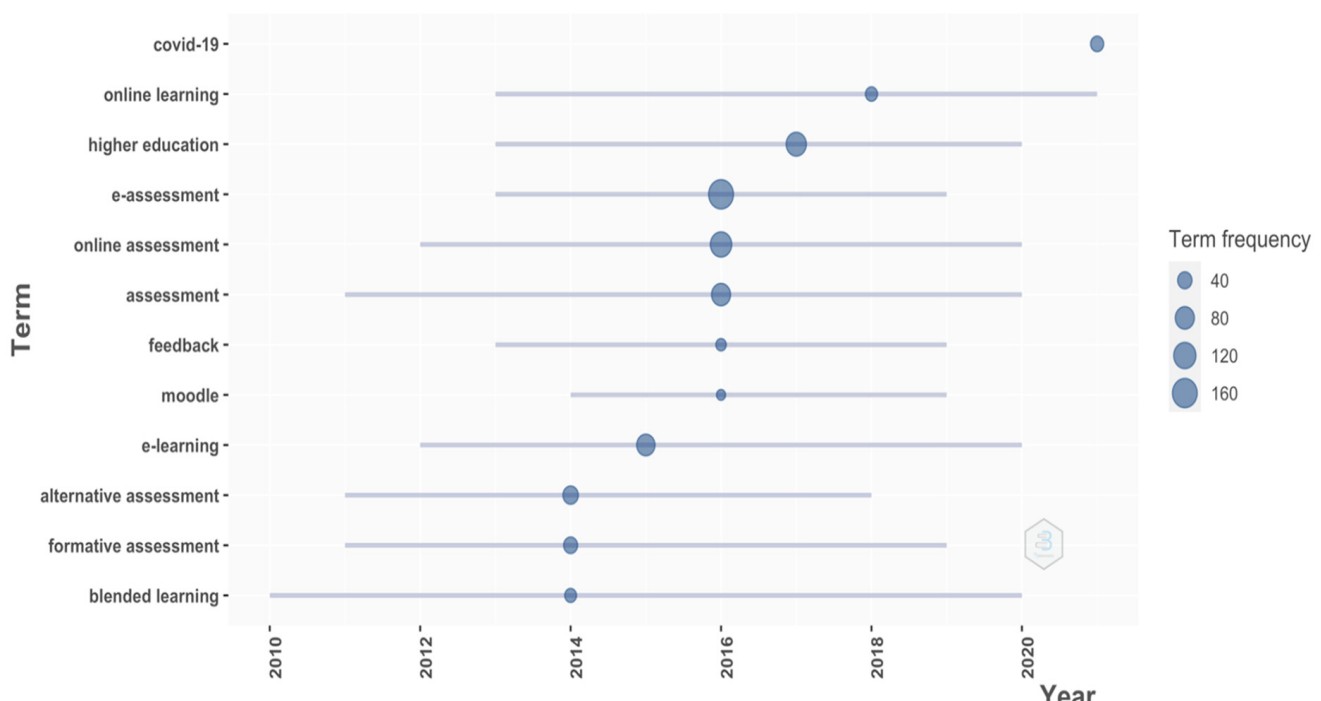

**Figure 8.** Trend topics.

According to Kessler [61], coupling refers to two articles that are said to be bibliographically coupled if at least one cited source or keyword appears in the reference or keyword lists of both articles. A network analysis approach based on the co-use of keywords added by the authors in the studies was applied. The concept of degree centrality was introduced, as well as weighted connections, which may be used to analyze co-authorship or citation

networks [62]. The impact dimension shows the level based on the frequency of use by the studies reviewed. As shown in Figure 9, three clusters have emerged. The red cluster contains "higher education, assessment, e-assessment, and e-learning" keywords, and it has a high impact but medium centrality value. In the red cluster, all studies have "higher education" keywords. In the green cluster, the words "e-assessment, e-learning, assessment, moodle" come to the fore. This group, on the other hand, has a high level of centrality but a lower impact value. Working volumes are high in the green cluster. In the green cluster, the percentage "e-assessment" keyword is 84.5. The blue cluster contains "online assessment, blended learning, e-learning and COVID-19". In this cluster, the centrality is low, but the impact value is medium. In this cluster, the highest percentage is 79.2 for the "online assessment" keyword.

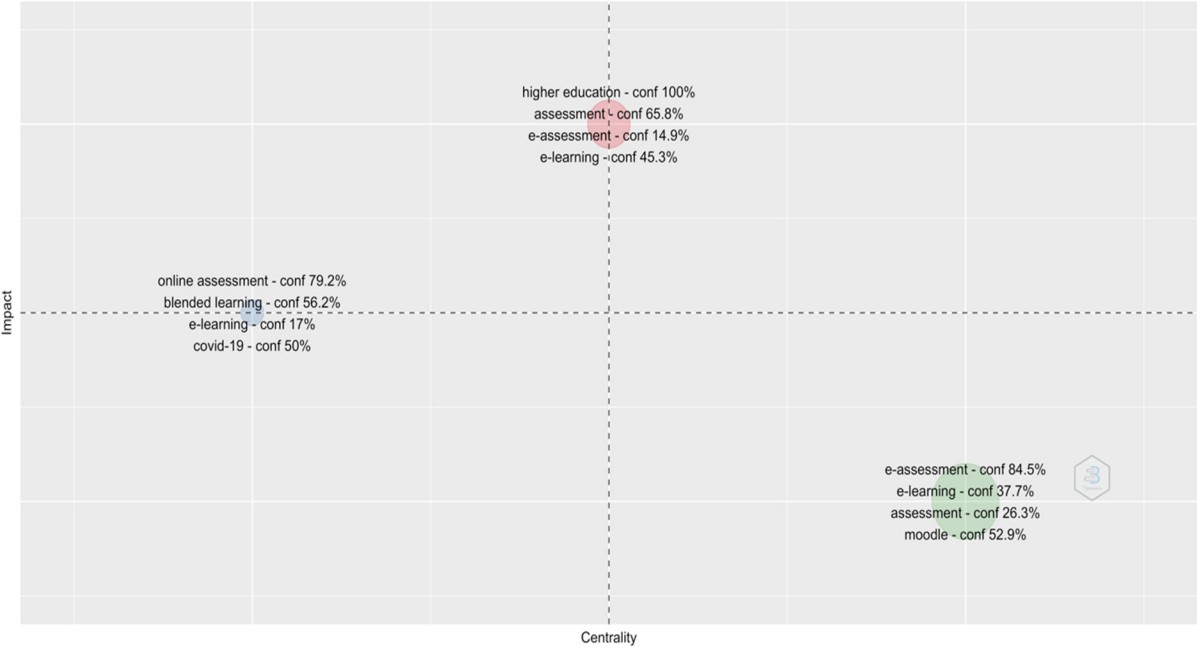

**Figure 9.** Clusters by Studies Coupling Based on keywords.

According to Callon et al. [63], the methodological foundation of co-word analysis is the idea that the co-occurrence of keywords describes the contents of the documents in a file. It may be viewed as a node in a larger network; one that is defined by its location, that is, by the collection of links connecting it to other clusters/nodes in the larger network. It may be viewed as a cluster of words that are connected to one another; it defines a more or less dense network that is more or less cohesive and resilient. In this study, a thematic map analysis based on the co-use of the authors' keywords was applied. Thematic networks are represented in two dimensions, with the axes representing the thematic network's centrality (the theme's relevance in the study area, horizontal axis) and density (a measure of the theme's development, vertical axis) [64]. Centrality measures the strength of a cluster's connections to other clusters. The more connections and the stronger they are, the more this cluster denotes a collection of critical research challenges identified by the scientific or technical community. Density refers to the strength of the linkages that connect the cluster's words. The more robust these connections are, the more the research challenges associated with the cluster form a cohesive and integrated whole. One may argue that density is an accurate depiction of a cluster's potential to persist and grow over time in the field under examination [63].

In accordance with the centrality and density of the research subjects, it is separated into four segments (As shown in Figure 10). In the first segment, the subjects in the left-bottom segment are the subjects that have decreased in density and centrality, which are referred to as "Declining Themes". In this segment, "electronic assessment", "computer-

based assessment", and "motivation" topics are seen. The subjects in the upper-left segment, on the other hand, are the subjects called "Niche Themes", which are subjects with high density and low centrality. Niche themes are sufficiently developed but have little relationship with other themes and studies. The subjects "alcohol", "college students", and "depression" formed a cluster. The working volume is low. On the other hand, the second cluster in this segment consists of "learning analytics", "e-assessment system", and "moocs" topics. The centrality of these issues is slightly higher. In the cluster, which has a medium level of centrality and a higher density, there are "online formative assessment", "collaborative learning", and "distance education". In the group with both centrality and high density, which is called "Motor themes", there are "alternative assessment", "self-assessment", and "authentic assessment" subjects. The fourth segment is called "basic themes". There are four themes in this segment. The theme, whose centrality is above medium, and its density is below medium, includes the subjects of "distance learning", "accessibility", and "assessment design". Another theme is "COVID", "medical education", and "online education" issues. The subjects of "online assessment", "formative assessment", and "online learning" are low in density but high in centrality. The number of studies carried out is in second place. The subjects with the highest centrality, that is, the most related to other themes, but the development of the theme within itself, that is, less intensity, were the subjects "e-assessment", "higher education", and "assessment".

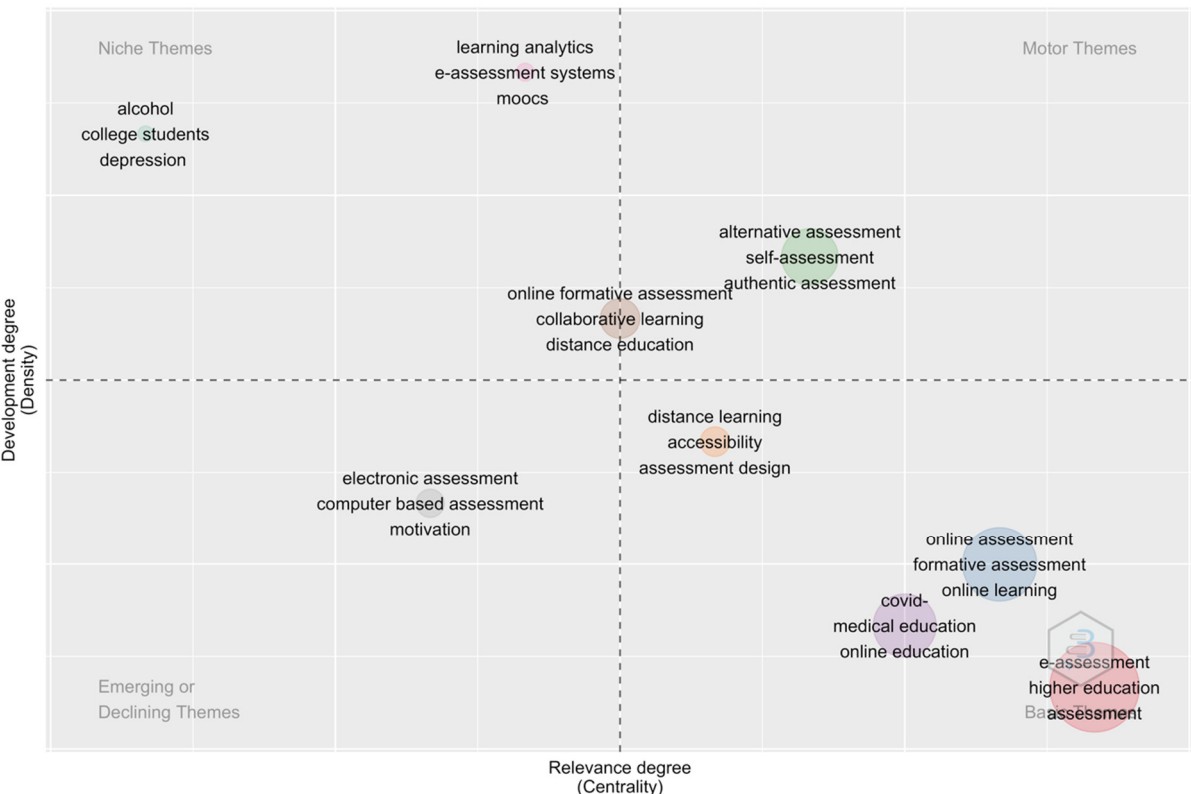

**Figure 10.** Clusters by Studies Themes.

The co-occurrences network is based on the analysis of the co-occurrence of keywords [39,63]. As shown in Figure 11, two clusters emerge. In the red cluster, the keywords "students", "education", "e-learning", and "teaching" come to the fore. The size of the circles indicates the frequency of the word, and the thickness of the lines indicates the frequency of use together. In the blue cluster, the keywords "human", "humans", "female", and "male" come to the fore.

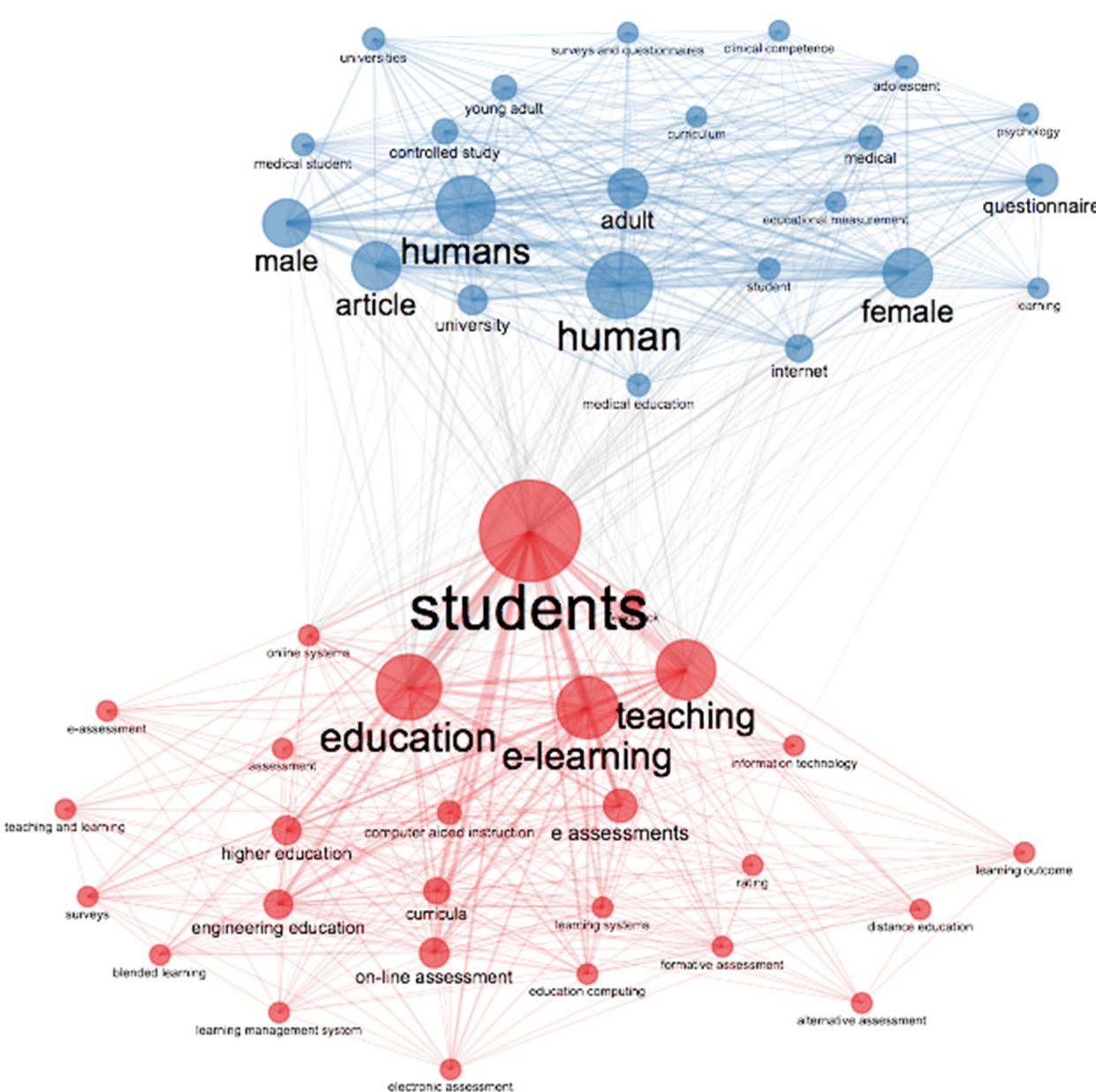

**Figure 11.** Co-occurrences network.

In the co-citation analysis part, the common publications that the studies used together in the reference lists were examined. The size of the circle indicates how often it is used, and the thickness of the lines indicates the frequency of use together. As a result of these analyses, four clusters emerged (As shown in Figure 12). Black and Willian [65] were the most famous studies in the red cluster. This study was conducted in conjunction with a review of the literature on formative assessment in the classroom. Another research examined the nature and purpose of formative evaluation in the process of expertise development [66]. Two studies [31,67] stand out in the blue cluster. Their articles include Scoping a vision for formative electronic assessment [31] on the nature of feedback [67]. The articles in the lilac cluster are titled "Online formative assessment in higher education: A literature review" [33] and "Formative assessment and self-regulated learning: A model and seven principles of effective feedback practice" [68]. The first two papers in the green cluster are titled "A review of the research on e-assessment" [53] and "e-assessment and the student learning experience: A survey of student perceptions of e-assessment" [47].

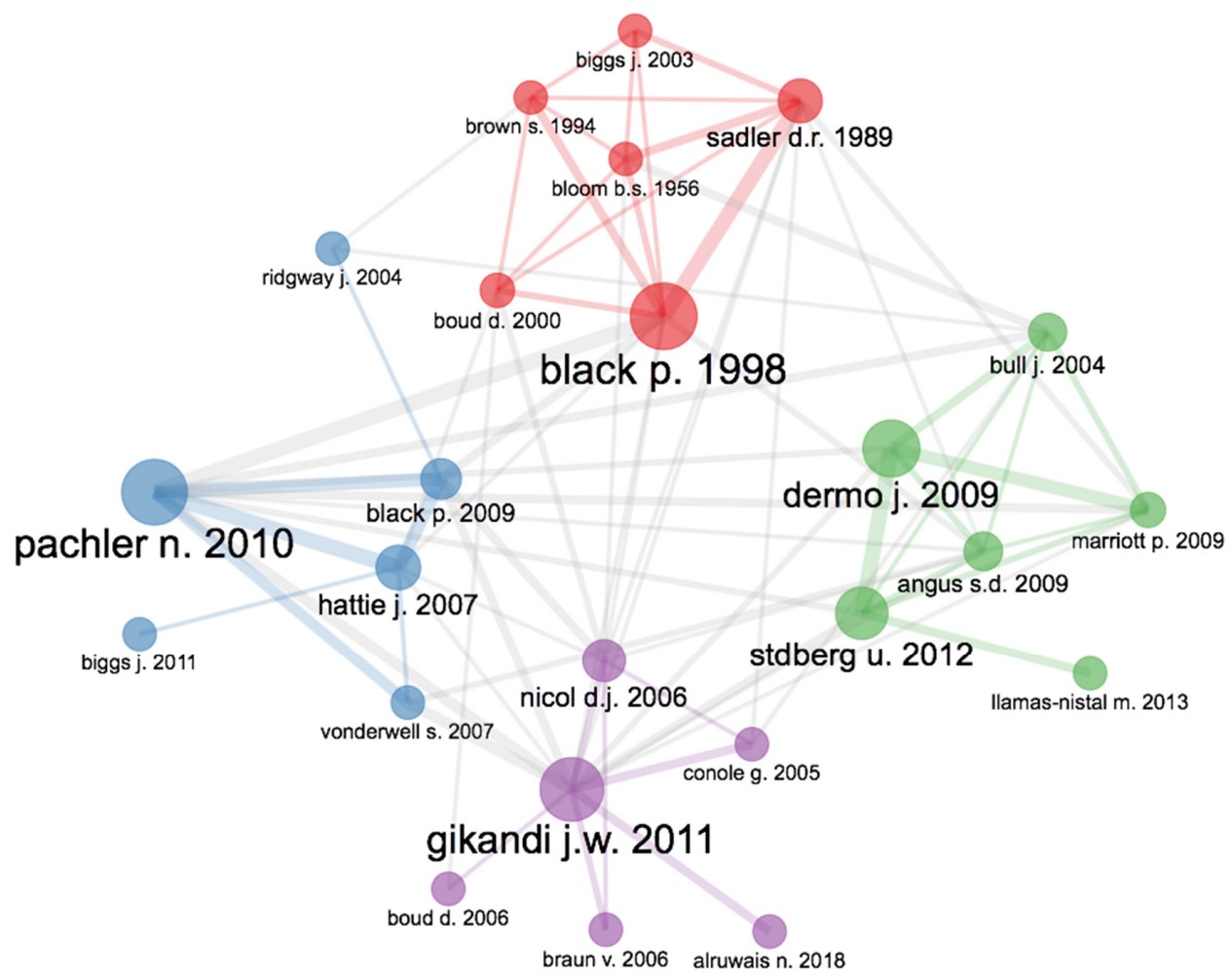

**Figure 12.** Co-citations network.

When the nations' participation in the collaborative endeavor is analyzed, five groups of countries are identified (As shown in Figure 13). According to the study's sample size, the United States, Canada, Japan, China, and Hong Kong are all included in the red cluster, with the United States taking the lead. The United Kingdom is in second position and is the leader of the blue cluster. The United Kingdom, Spain, Turkey, Ireland, Bulgaria, the Netherlands, Mexico, and Italy are among the countries represented in this category. Finland and Portugal, on the other hand, are included in the orange cluster, and Finland is also related to the blue cluster. In a similar vein, Germany, which belongs to the lilac group, is linked to the blue group. Germany, Austria, Switzerland, and Belgium are among the countries that belong to the lilac group. The green cluster, on the other hand, comprises countries such as Australia, Indonesia, and Malaysia. A shared language and geographical closeness between the nations are expected to contribute to greater collaboration between the two countries. Some countries (for example, Indonesia, Malaysia, and Hong Kong) prefer more cooperation only in their geographical regions, whereas some countries (such as USA and UK) play a central role in cooperation.

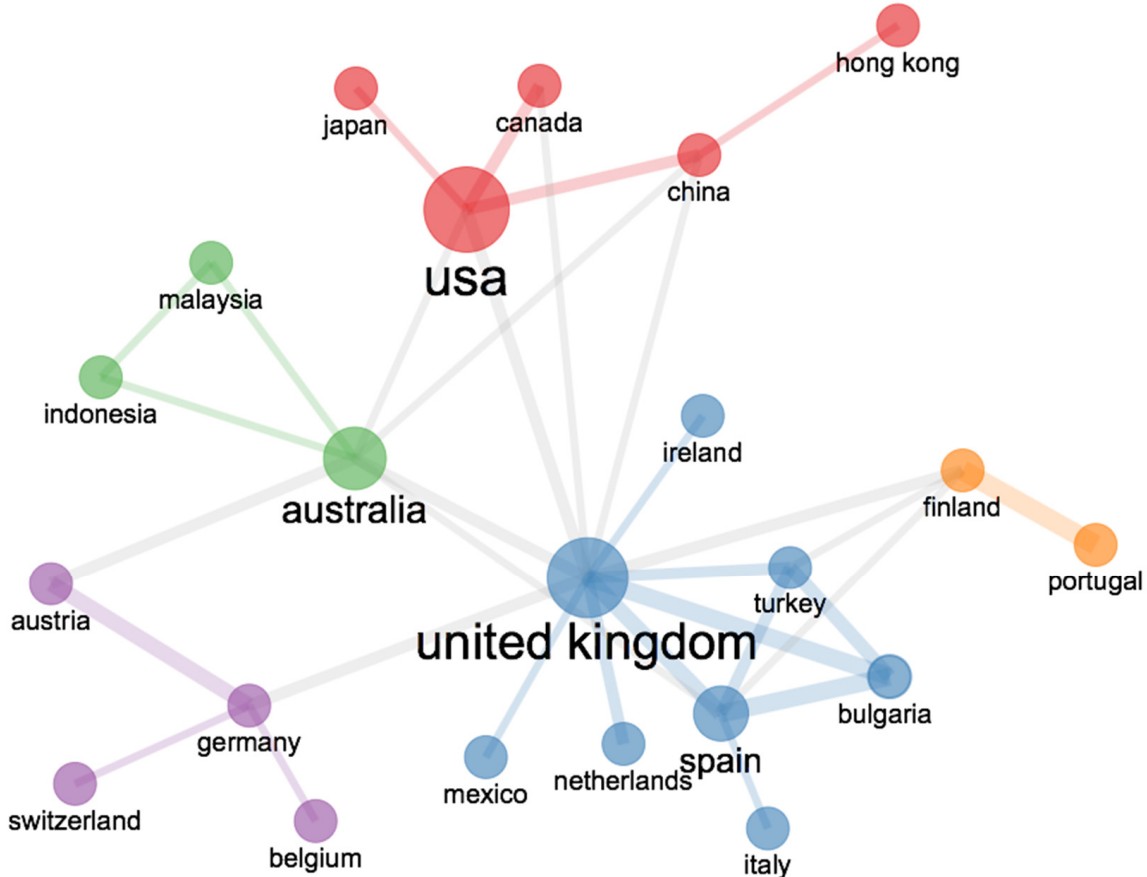

**Figure 13.** Country participations network.

## 4. Conclusions

A bibliometric analysis was conducted regarding the use of online formative assessment in higher education. In the study, studies in the Scopus database were searched with keywords. In total, 898 studies were analyzed. According to the study results, there were 0.382 documents per author, and there were 2.62 authors per document. There were 2.97 co-authors per document, and the collaboration index was 3.02. The annual growth rate was 17.44%. The rate of increase between the years 2019 and 2021 was higher than in other years. The average number of citations per document by year was 2.56. The most effect sources were *Assessment & Evaluation in Higher Education* and *Computers & Education*.

According to total citations, first place was Gikandi et al. [33], but after normalization, [56] will be the first rank. The most effective authors were Gikandi Jw according to total citations; according to h-index, Vrlan Gm; g -index, Guerrero-Roldn Ae; and m-index, Garca-Pealvo Fj. Schroeder U has 10 years of production, from 2010 to 2019. Then, Babo R and Garca-Pealvo FJ have 9 years. According to SCP and MCP, the USA is highest according to SCP, but the UK is highest in MCP, and Turkey, South Africa, and India have only SCP. When the effects of the institutions are examined, the most effective ones are RWTH Aachen University and Universitat Oberta De Catalunya.

Between 2010 and 2020, the phrase "blended learning" gained widespread popularity. Between 2011 and 2019, the words formative evaluation and assessment were often employed. Between 2015 and 2019, the authors used the term "moodle". In 2020, the term "COVID" became widely used in publications. Based on coupling keywords, three groupings formed. The red cluster contains keywords related to higher education, assessment, e-assessment, and e-learning. The green cluster has the phrases "e-assessment, e-learning, assessment, moodle". This group has high centrality but low influence. The green cluster has a lot of studies. The percentage of "e-assessment" is 84.5 in the green cluster. On-line

assessment, blended learning, e-learning, and COVID-19 are all in the blue cluster. This cluster has low centrality but high effect. The highest percentage for "online assessment" is 79.2.

Themes were divided into four parts based on the centrality and density of the study topics. Declining themes were electronic, computer-based, and motivational subjects. Niche subjects had minimal relevance to broader themes and disciplines. "alcohol", "college students", and "depression" clustered. The second cluster in this area included themes such as "learning analytics", "e-assessment system", and "moocs". These concerns were significantly more central. The cluster also contained "online formative assessment", "collaborative learning", and "distance education". The section on "Motor themes" included topics such as "alternative assessment", "self-assessment", and "authentic assessment". "Basic themes" were "distance learning", "accessibility", and "assessment design". The second one was "Covid", "medical education", and "online education". The third one was "online assessment", "formative assessment", and "online learning", which are less dense but more critical terms. The subjects "e-assessment", "higher education", and "assessment" had the greatest centrality, meaning they were the most related to other themes but had the least intensity within the theme.

According to the co-occurrence network analysis, two clusters developed. In the red cluster, students, education, e-learning, and teaching were prominent. The circles' sizes denoted the words' frequency, whereas the lines' thicknesses showed their combined frequency. The terms "human", "humans", "female", and "male" dominated the blue cluster. According to a co-citation analysis, four clusters emerged. Analyzing the countries' engagement in the coordinated effort revealed five groupings. The red cluster included the United States, Canada, Japan, China, and Hong Kong, with the United States leading the way. The UK ranked second and led the blue cluster. This group included the UK, Spain, Turkey, Ireland, Bulgaria, the Netherlands, Mexico, and Italy. Finland and Portugal were in the orange cluster, whereas Finland was also in the blue cluster. Similarly, the lilac group's Germany was related to the blue group. The lilac group included Germany, Austria, Switzerland, and Belgium. The green cluster included nations such as Australia, Indonesia, and Malaysia. The nations' shared language and physical proximity perhaps fostered more coordination.

This study, like previous bibliometric studies, gave insight into online formative assessment, provided a prognosis for future studies, and revealed cooperative potentials by analyzing historical study data. The most significant limitation of the study is that it only included studies that were indexed by the Scopus database. There might have been studies that contributed significantly to the field of OFA but were not indexed by Scopus; therefore, these studies were not available. The analyses, on the other hand, were carried out in accordance with the keywords selected by the authors. If you use alternative terms that are developed in a broader context, you might obtain different outcomes. When performing bibliometric studies on this issue, researchers will be able to provide a more comprehensive analysis by merging the studies that will be collected from multiple databases. Researchers interested in working on OFA can also perform studies concentrating on subjects such as "alternative assessment", "self-assessment", and "authentic assessment" within the framework of motor themes, according to their interests.

**Author Contributions:** All authors have contributed equally. All authors have read and agreed to the published version of the manuscript.

**Funding:** This research received no external funding.

**Institutional Review Board Statement:** Not applicable.

**Informed Consent Statement:** Not applicable.

**Data Availability Statement:** The data that support the findings of this study are available from the corresponding author, upon reasonable request.

**Conflicts of Interest:** The authors declare no conflict of interest.

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
