# Peer review of "Online Formative Assessment in Higher Education: Bibliometric Analysis"

_education, doi:10.3390/educsci12030209_

Round 1

Reviewer 1 Report

If possible, expand on the discussion section by adding few examples from the Asian/south-Asian higher education context.

Author Response

reviewer 1: If possible, expand on the discussion section by adding few examples from the Asian/south-Asian higher education context.

Respond: Revisions were made in the discussion section, but since only cooperation analysis was made on geographical issues, no specific examples were given.

Reviewer 2: We appreciate that is interesting research that might have a meaningful contribution to the field.

Suggestions for Authors:

substantiate and add the research questions; add some information about The Scopus database (argument why this database was selected to be used in your research);

R: Research questions added. Added information about the Scopus database. The justification has been written.

specify the source of the figures (author's source..)

check the MDPI references guide and cite the references in the text accordingly  -  mdpi_references_guide_v5.pdf (mdpi-res.com)

R: Necessary revisions have been made.

discuss the research results in the light of other studies and research; are your research results in accordance with the results of other studies?

R: The discussion section has been revised. No bibliometric study was found on online formative assessment in higher education, but connections were made with systematic literature review studies related to the topic.

reorganize the Conclusion section; some parts of it should be presented in a separate Results discussion section;

formulate clearly the conclusions, research limitations, main contributions highlight better your  contribution to the literature and the usefulness of your research

R: Necessary revisions have been made.

Reviewer 3: In this article, a bibliometric study of the scientific literature related to formative evaluation and its online evolution during the COVID stage is carried out.

It is an interesting work, although there are some minor considerations that would improve its readability and interest for the educational community.

It would be desirable to clearly define the objective set out in the study which, although it is developed incrementally throughout the work, a proper early definition would facilitate reading and interest.

R: Necessary revisions have been made. Research questions added.

In addition, Figure 1 should be replaced by the flow chart typical of the PRISMA methodology.

R: Necessary revisions have been made.

The Bibliometrix tool used is adequate, but it would be desirable that, together with the result, a specific interpretation of what each table and figure means and represents be made.

R: Necessary revisions have been made.

The meaning of an acronym should be defined in the first quotation of the text. Sometimes the meaning is in the figure itself and this makes it difficult to read the text linearly.

R: Necessary revisions have been made.

The conclusions need an improved wording that groups the thematic blocks, highlights the contributions obtained in the work and compares them with other published knowledge on the subject formative evaluation and its online evolution.

R: Conclusion section is revised.  No bibliometric study was found on online formative assessment in higher education, but connections were made with systematic literature review studies related to the topic in discussion section.

Reviewer 2 Report

In this article, a bibliometric study of the scientific literature related to formative evaluation and its online evolution during the COVID stage is carried out.
It is an interesting work, although there are some minor considerations that would improve its readability and interest for the educational community.
It would be desirable to clearly define the objective set out in the study which, although it is developed incrementally throughout the work, a proper early definition would facilitate reading and interest.
In addition, Figure 1 should be replaced by the flow chart typical of the PRISMA methodology.
The Bibliometrix tool used is adequate, but it would be desirable that, together with the result, a specific interpretation of what each table and figure means and represents be made.
The meaning of an acronym should be defined in the first quotation of the text. Sometimes the meaning is in the figure itself and this makes it difficult to read the text linearly.

The conclusions need an improved wording that groups the thematic blocks, highlights the contributions obtained in the work and compares them with other published knowledge on the subject formative evaluation and its online evolution.

Author Response

Reviewer 3: In this article, a bibliometric study of the scientific literature related to formative evaluation and its online evolution during the COVID stage is carried out.

It is an interesting work, although there are some minor considerations that would improve its readability and interest for the educational community.

It would be desirable to clearly define the objective set out in the study which, although it is developed incrementally throughout the work, a proper early definition would facilitate reading and interest.

R: Necessary revisions have been made. Research questions added.

In addition, Figure 1 should be replaced by the flow chart typical of the PRISMA methodology.

R: Necessary revisions have been made.

The Bibliometrix tool used is adequate, but it would be desirable that, together with the result, a specific interpretation of what each table and figure means and represents be made.

R: Necessary revisions have been made.

The meaning of an acronym should be defined in the first quotation of the text. Sometimes the meaning is in the figure itself and this makes it difficult to read the text linearly.

R: Necessary revisions have been made.

The conclusions need an improved wording that groups the thematic blocks, highlights the contributions obtained in the work and compares them with other published knowledge on the subject formative evaluation and its online evolution.

R: Conclusion section is revised.  No bibliometric study was found on online formative assessment in higher education, but connections were made with systematic literature review studies related to the topic in discussion section.

Reviewer 3 Report

We appreciate that is interesting research that might have a meaningful contribution to the field.

Suggestions for Authors:

  • substantiate and add the research questions;
  • add some information about The Scopus database (argument why this database was selected to be used in your research);
  • specify the source of the figures (author's source..)
  • check the MDPI references guide and cite the references in the text accordingly  -  mdpi_references_guide_v5.pdf (mdpi-res.com)
  • discuss the research results in the light of other studies and research; are your research results in accordance with the results of other studies? 
  • reorganize the Conclusion section; some parts of it should be presented in a separate Results discussion section;
  • formulate clearly the conclusions, research limitations, main contributions
  • highlight better your  contribution to the literature and the usefulness of your research

Author Response

Reviewer 2: We appreciate that is interesting research that might have a meaningful contribution to the field.

Suggestions for Authors:

substantiate and add the research questions; add some information about The Scopus database (argument why this database was selected to be used in your research);

R: Research questions added. Added information about the Scopus database. The justification has been written.

specify the source of the figures (author's source..)

check the MDPI references guide and cite the references in the text accordingly  -  mdpi_references_guide_v5.pdf (mdpi-res.com)

R: Necessary revisions have been made.

discuss the research results in the light of other studies and research; are your research results in accordance with the results of other studies?

R: The discussion section has been revised. No bibliometric study was found on online formative assessment in higher education, but connections were made with systematic literature review studies related to the topic.

reorganize the Conclusion section; some parts of it should be presented in a separate Results discussion section;

formulate clearly the conclusions, research limitations, main contributions highlight better your  contribution to the literature and the usefulness of your research

R: Necessary revisions have been made.

Round 2

Reviewer 3 Report

 Accept in present form